Morphological variation of Aphidius ervi Haliday (Hymenoptera: Braconidae) associated with different aphid hosts

Villegas Cinthya M. 1
Žikić Vladimir 2
Stanković Saša S. 2
Ortiz-Martínez Sebastián A. 1
Peñalver-Cruz Ainara ainara.penalver@gmail.com 1
Lavandero Blas blavandero@utalca.cl 1
1 Laboratorio de Interacciones Insecto-Planta, Instituto de Ciencias Biológicas, Universidad de Talca , Talca , Chile
2 Department of Biology and Ecology, Faculty of Science and Mathematics, University of Niš , Niš , Serbia
Huber Dezene
Electronic publication date: 2017 Jul 11
Publication date: 2017
Volume: 5
Electronic Location ID: e3559
Received 2017 Mar 1; Accepted 2017 Jun 19
Copyright: ©2017 Villegas et al.
Copyright year: 2017
Copyright holder: Villegas et al.
License: This is an open access article distributed under the terms of the Creative Commons Attribution License, which permits unrestricted use, distribution, reproduction and adaptation in any medium and for any purpose provided that it is properly attributed. For attribution, the original author(s), title, publication source (PeerJ) and either DOI or URL of the article must be cited.
License URL: https://creativecommons.org/licenses/by/4.0/

Keywords: Variability, Biotypes, Geometric morphometrics, Allometry, Wing shape, Wing size, Parasitoids

Funding: Fondecyt Grant 1110341 1140632 Marie Curie Actions 611810-Aphidweb BECA Guillermo Blanco Postdoctoral Fondecyt Grant 3160233 The Ministry of Education, Science and Technological Development of the Republic of Serbia III43001 Research was funded by Fondecyt Grant No 1110341, 1140632 and Marie Curie Actions 611810-Aphidweb. CV was supported by BECA Guillermo Blanco during her Master thesis. AP was supported by Postdoctoral Fondecyt Grant 3160233. This research was also partially funded by the Grant III43001 (The Ministry of Education, Science and Technological Development of the Republic of Serbia). There was no additional external funding received for this study. The funders had no role in study design, data collection and analysis, decision to publish, or preparation of the manuscript.

==============================
Background

Parasitoids are frequently used in biological control due to the fact that they are considered host specific and highly efficient at attacking their hosts. As they spend a significant part of their life cycle within their hosts, feeding habits and life history of their host can promote specialization via host-race formation (sequential radiation). The specialized host races from different hosts can vary morphologically, behaviorally and genetically. However, these variations are sometimes inconspicuous and require more powerful tools in order to detect variation such as geometric morphometrics analysis.

Methods

We examined Aphidius ervi, an important introduced biological control agent in Chile associated with a great number of aphid species, which are exploiting different plant hosts and habitats. Several combinations (biotypes) of parasitoids with various aphid/host plant combinations were analyzed in order to obtain measures of forewing shape and size. To show the differences among defined biotypes, we chose 13 specific landmarks on each individual parasitoid wing. The analysis of allometric variation calculated in wing shape and size over centroid size (CS), revealed the allometric changes among biotypes collected from different hosts. To show all differences in shape of forewings, we made seven biotype pairs using an outline-based geometric morphometrics comparison.

Results

The biotype A. pis_pea (Acyrthosiphon pisum on pea) was the extreme wing size in this study compared to the other analyzed biotypes. Aphid hosts have a significant influence in the morphological differentiation of the parasitoid forewing, splitting biotypes in two groups. The first group consisted of biotypes connected with Acyrthosiphon pisum on legumes, while the second group is composed of biotypes connected with aphids attacking cereals, with the exception of the R. pad_wheat (Rhopalosiphum padi on wheat) biotype. There was no significant effect of plant species on parasitoid wing size and shape.

Discussion

Although previous studies have suggested that the genotype of parasitoids is of greater significance for the morphological variations of size and shape of wings, this study indicates that the aphid host on which A. ervi develops is the main factor to alter the structure of parasitoid forewings. Bigger aphid hosts implied longer and broader forewings of A. ervi.

Introduction

Parasitoids are frequently used in biological control as they are considered to be highly specialized natural enemies (Godfray, 1994). By being highly specialized, released parasitoids will be the most efficient at attacking the target pest species. This reduces the possibility of environmental harm of rapidly-growing parasitoid populations migrating from crops into adjacent natural habitats (Rand, Tylianakis & Tscharntke, 2006), as has been observed for generalist predators (Duelli et al., 1990; French et al., 2001). Although several parasitoid species can exploit many hosts (Mackauer & Starý, 1967) this may not be consistent across an entire species, and different biotypes may be specialized to different hosts/environments (Stireman et al., 2006; Forbes et al., 2009). Previous studies have shown that host-associated biotypes of parasitoids from different hosts/environments can vary morphologically, behaviorally and genetically (Žikić et al., 2009; Feder & Forbes, 2010; Kos et al., 2012; Zepeda-Paulo et al., 2013). In terms of morphological features, the shape and size of their appendages have shown great promise for separating host-associated races of parasitoids. Among these, insect wings are especially relevant as they are two dimensional structures with important characteristics, in terms of adaptation and function (Wootton, 2002; Žikić et al., 2009). Previous studies have shown that the size, shape and venation of the wings can be important features to separate species and characterize populations within a single species (Sadeghi, Adriaens & Dumont, 2009). A geometric morphometrics approach is very useful for detecting minute variations in morphology of different parasitoid populations which otherwise cannot be identified easily (Villemant, Simbolotti & Kenis, 2007; Žikić et al., 2009; Kos et al., 2011). This can be of great importance because these morphological variations in wing shape could be associated with a specific environment or host-associated population of a parasitoid species.

The Chilean populations of Aphidius ervi (Haliday, 1834) (Hymenoptera: Braconidae) may be a good example where different host associations and environment have influenced morphology. This species is an oligophagous parasitoid associated with a number of legumes, Solanaceae and cereal aphid species. Legume feeding aphid hosts include Acyrthosiphon pisum (Harris, 1776), Acyrthosiphon kondoi (Shinji, 1938) and Macrosiphum euphorbiae (Thomas, 1878) with Aulacorthum solani (Kaltenbach, 1843) feeding on Solanaceae (Takada & Tada, 2000). Cereal aphid hosts include Sitobion avenae (Fabricius, 1775), Rhopalosiphum padi (Linnaeus, 1758), Schizaphis graminum (Rondani, 1852) and Metopolophium dirhodum (Walker, 1849) (Starý, 1993). Aphidius ervi was introduced in Chile in the 1970’s as part of a classical biological control strategy to minimize the damage caused by the grain aphid (S. avenae) on cereals and maintain the pest population at low densities in the field (Zúñiga et al., 1986). Currently, A. ervi is the predominant parasitoid species controlling A. pisum and S. avenae. It represents more than 94% of parasitized A. pisum on legumes and 38% of parasitized S. avenae on cereals and is considered a highly efficient biological control agent of aphids on both crops (Gerding et al., 1989; Starý et al., 1994; Zepeda-Paulo et al., 2013). The main goal of the present study is to analyze the shape and size of forewings of A. ervi collected in different plant/host associations, on legumes and cereals.

Materials & Methods

Sampled material

Aphids were collected from fields of legumes and cereals in two different geographic regions of central Chile: “Región de los Rios” (S39°51′, W73°7′) and “Región del Maule” (S35°24′, W71°40′). Parasitoids were obtained from parasitized aphids collected in the field, and after emergence carefully examined and identified. Reared samples were transferred in the growing laboratory and treated under following conditions: 20 °C, 50–60% RH, D16:N8 of photoperiod. Parasitoid wasps were put in plastic microtubes with 96% ethyl alcohol. Paraisitoid identification followed Starý (1995) for the taxonomical identification.

A total of 131 females of Aphidius ervi were analyzed. Parasitoids were divided into eight biotypes according to their aphid hosts and to the plant species where the aphids were found (Table 1). The biotypes used for Acyrthosiphon pisum were the alfalfa biotype from alfalfa (Medicago sativa L.), the pea biotype from pea (Pisum sativum L.), and the clover biotype from red clover (Trifolium pratense L.). Biotypes reared on cereals were the bird cherry-oat aphid (Rhopalosiphum padi), the rose grain aphid (Metopolophium dirhodum) the green-bug (Schizaphis graminum), and the grain aphid (Sitobion avenae) sampled from wheat (Triticum aestivum L.). Another cereal biotype is also the grain aphid (Sitobion avenae) which was collected from oat (Avena sativa L.) (Table 1).

Table 1 Aphidius ervi material sampled and biotype definitions.

Aphid host	Host-plant	No of specimens	Biotype	
Acyrthosiphon pisum	alfalfa	29	A. pis_alfalfa	
Acyrthosiphon pisum	pea	28	A. pis_pea	
Acyrthosiphon pisum	red clover	14	A. pis_clover	
Metopolophium dirhodum	wheat	10	M. dir_wheat	
Rhopalosiphum padi	wheat	10	R. pad_wheat	
Schizaphis graminum	wheat	13	Sc. gra_wheat	
Sitobion avenae	oat	14	S. ave_oat	
Sitobion avenae	wheat	13	S. ave_wheat	
Total		131		

Geometric morphometrics

To conduct the geometric morphometrics analysis, we applied two-dimensional landmark-based methods (Bookstein, 1986; Bookstein, 1991). Right forewings of each female parasitoid were removed and mounted in Neo Mount (Merck) following the procedure described in Žikić et al. (2009). Forewings were recorded using an OPTIKA SZN (45x) stereoscopic compound microscope with a mounted 5-megapixel photographic camera using software Optika Vision Pro v2.7. Using the geometric morphometrics method (Zelditch et al., 2004) we determined and quantified morphological variations of wing size and shape in different Aphidius ervi biotypes.

Figure 1 Right forewing of Aphidius ervi; set of 13 specific landmarks.

Table 2 Description of specific forewing landmarks used in the analyses. Wing vein terminology follows Wharton, Sharkey & Michael (1997).

Landmark number	Landmark definition	
1	Beginning of stigma	
2	Corner at the middle of stigma and r vein	
3	End of stigma	
4	End of metacarpus	
5	Projection of RS vein on the edge of wing	
6	Projection of M vein on the edge of wing	
7	Projection of CU vein on the edge of wing	
8	Corner of RS and r-m veins	
9	Corner of M and r-m veins	
10	Corner of m-cu and 1CU veins	
11	Corner of 1CU and 1A veins	
12	Corner of 1M and 1CU	
13	Beginning of parastigma	

Eight different aphid-host/plant-host associations were used for morphological characterization of A. ervi biotypes (Table 1). To analyze the variation in wing shape of parasitoids, 13 specific landmarks were scored for each forewing. Positioned landmarks were digitized using software TpsDig v2.16 (Rohlf, 2010) (Fig. 1 and Table 2). Using a generalized procrustes analysis (an analysis that allows morphotype separation) all variations due to scale, orientation and position of the 13 landmark configurations were eliminated (Rohlf & Slice, 1990; Bookstein, 1991). Centroid size (CS) was calculated for each forewing, indicating the dispersion of the landmarks from the centroid; this parameter is used as a relative indicator of the wing size. Size variation among forewings (obtained on the basis of the CS) was examined using the analysis of variance (ANOVA) performed on the centroid size. To see if there were some correlations between the wing size and shape, we performed a regression test between the CS and procrustes coordinates (PC) scores (Žikić et al., 2010). Discriminant analysis using the residuals of the regression test was performed to determine if any of the procrustes distances were statistically significant. The latter to understand if changes in wing shape were caused by changes of the wing size. Resultant shape variables were also analyzed using multivariate analysis of variance (MANOVA) performed on eigenvalues of the PC scores. The MorphoJ software was used to analyze and visualize shape changes described by canonical axes (Klingenberg, 2011). Principal component analysis (PCA) was used to analyze variability in wing shape among the specimens investigated. This analysis allowed us to group the different biotypes studied. The differences in wing shape were visualized using canonical variate analysis (CVA) in order to observe the variability among the A. ervi biotypes (Rohlf, 2010) (Fig. S2). The centroid sizes were obtained using the MorphoJ v1.06b software (Klingenberg, 2011). For the visualization of wing shape changes between the analysed biotypes, outline drawings consisting of a series of lines that are in a specific relation to the arrangement of the landmarks were created. MorphoJ uses the thin-plate spline method to produce a deformation of the drawing so that the arrangement of landmark points matches the configurations that are to be visualized (see Klingenberg, 2011). All statistical tests concerning analysis of variance (ANOVA) and multivariate analysis of variance (MANOVA) were performed in Statistica 7.0 (StatSoft, Tulsa, OK, USA).

Results

Significant differences in shape were observed with the procustes ANOVA analyses (F = 17.30; df = 7; P < 0.000001). However, according to the PCA, the variability explained by the first three axes was rather low; all three explain 50.6% of the total variability (Fig. S1). Forewing size, and shape were significantly different using the PC scores (MANOVA: Wilks’ λ = 0.112737; F = 1.74; df = 154; P < 0.000001). Considering that all statistical tests of variance were statistically significant, we performed a canonical variate analysis (CVA) to observe the variability among the A. ervi biotypes (Fig. S2). However, there was no conspicuous grouping of the biotypes into discrete morphotypes. The first canonical axis (CV1) explains 38.4%, while the second axis (CV2) explains only 23% of the total variability. To see if there was some correlation between the wing size and shape we performed the regression test between the centroid size and PC scores. The statistical test showed that the wing shape is clearly correlated with the wing size (P-value: <0.0001; Fig. 2). The percentage of the wing shape variability explained by this regression test is only 6.78% (% predicted: 6.7783%), therefore the wing size has a small contribution to variations in wing shape. The largest wings were of the specimens from the biotype A. pis_pea, while the smallest were those from A. ervi parasitizing S. avenae on wheat (biotype S. ave_wheat) and on S. graminum also on wheat (Fig. 2).

Figure 2 (A): Figure of forewing shape changes in A_pis_p biotype. The blue line represents the largest wing shape analyzed, while the gray line represents the average wing shape. (B): The regression results of the centroid size (CS) and PC scores (permutation test against the null hypothesis of independence, P-value: <0.0001).

The included biotypes were Acyrthosiphon pisum from alfalfa (A_pis_a), A. pisum from red clover (A_pis_c), A. pisum from pea (A_pis_p), Metopolophium dirhodum from wheat (M_dir_w), Rhopalosiphum padi from wheat (R_pad_w), Sitobion avenae from oat (S_ave_o) and wheat (S_ave_w) and Schizaphis graminum from wheat (S_gra_w).

Considering that the regression result was statistically significant (P-value: <0.0001) we performed a discriminant analysis (DA) using the residuals to clarify the influence of the wing size on its shape. This particular analysis showed that none of the procrustes distances were statistically significant (P-value: >0.05), suggesting that although small there are some morphological changes caused by the variation in size. Given that the biotype A. pis_pea has the largest wings, we wanted to visualize how the wings of all other A. ervi biotypes change in relation to this particular biotype (A. pis_pea) using an outline-based geometric morphometric method (Fig. 3). The changes between the biotype A. pis_pea and the other six can be seen in Fig. 3.

Figure 3 Outline-based comparison of the wing shape between the biotype A. pis_pea and the rest of the seven biotypes: Acyrthosiphon pisum wing shape from pea (A. pis_pea) was compared to A. pisum from alfalfa (A. pis_alfalfa) (A), to A. pisum from red clover (A. pis_clover) (B), to Metopolophium dirhodum from wheat (M. dir_wheat) (C), to Rhopalosiphum padi from wheat (R. pad_wheat) (D), to Sitobion avenae from oat (S. ave_oat) (E), to Sitobion avenae from wheat (S. ave_wheat) (F) and to Schizaphis graminum from wheat (S. gra_wheat) (G).

Shape differences are the results of discriminant analysis (DA). The scale factor is increased by 5. The grey outline represents the biotype A. pis_pea; the black outline represents the other biotypes compared.

The least observed changes of the wing shape were detected between the following pairs: A. pis_pea/A. pis_alfalfa, A. pis_pea/A. pis_clover and A. pis_pea/R. pad_wheat (see relations in Fig. 3 and Fig. S2). More conspicuous changes were visible for the comparison between A. pis_pea/S. ave_oat, and A. pis_pea/S. ave_wheat. The latter changes are due to the narrowing of the wing in the two biotypes (S. ave_oat and S. ave_wheat). The greatest difference observed was between the biotype A. pis_pea and S. gra_wheat: this biotype has the narrowest wing in relation to A. pis_pea (Fig. 3 and Fig. S2).

Discussion

Aphidius ervi is known to attack economically important pests worldwide and in the Chilean agricultural landscapes it is considered a successful example of classical biological control of legume and cereal aphids (Starý, 1993; Starý et al., 1993; Rojas, 2005). Although it is very efficient in parasitizing target aphid pests, it has not been observed attacking native aphid species in shared environments (e.g: Uroleucon species developing on native plants in and around agricultural valleys in Chile) (Zúñiga et al., 1986; Starý, 1993). Many studies have shown heritable host fidelity and have hypothesized the possibility of different host associated biotypes. However, recent studies of Bilodeau et al. (2013) and Zepeda-Paulo et al. (2013) using population genetics suggest that in both North America and Chile there are no specialized races or biotypes on different aphid-host species, revealing high gene flow between these parasitoid populations.

In a recent study, it has been shown that the parasitoid genotype can have a stronger influence on wing shape than developing on a different parasitoid host species (Parreño et al., 2016). These authors used five asexual lines of Lysiphlebus fabarum (Marshall, 1896) (Braconidae) and four aphid hosts, and they found by using the procrustes coordinates on wings that the lineages acted as a better grouping factor compared to the parasitoid aphid-host variable. In this study, we did not discover any distinctive morphological features that could differentiate the Chilean populations of A. ervi. However, the significant narrowing of the wings observed for the S. ave_wheat and S. gra_wheat biotypes when compared to the A. pis_pea biotype is an indication of environmental and ecological effects particular to each parasitoid population (Fig. 3). The low genetic variability observed between specimens of A. ervi from different aphid host and locations suggests a high gene flow between parasitoid populations, with the result of no local adaptation or host associated races (Zepeda-Paulo et al., 2016).

Comparing the allometric relationships of wings among tested biotypes, it was found that the smallest wings were from S. gra_wheat, while the biggest wings were from A. pis_pea biotype (Fig. 2). This particular variability in wing size has morphological effects on the wing shape, causing the subtle changes among analyzed biotypes (Fig. 3). Therefore, this particular wing from the A. pis_pea biotype was used to compare it with the wings of the other seven biotypes (Fig. 3).

Conspicuous differences of the wing size and shape between A. pis_pea and other biotypes were clearer for those biotypes reared on cereals, compared to those biotypes from legumes. The specimens of this particular biotype have generally larger forewings than the other biotypes and are broader in the middle and the distal part (Figs. 2 and 3). The least deviation from the average wing constructed is observed for the R. pad_wheat biotype, where the differences were less noticeable (Fig. 3). This could be the effect of the aphid host size, because Acyrthosiphon pisum is rather a large aphid in comparison to Rhopalosiphum padi. Certainly, the biotypes reared from Acyrthosiphon pisum (A. pis_alfalfa, A. pis_clover and A. pis_pea) have the largest wings independent of the aphid clone (host-plant). Compared to all other analyzed aphid species (≤3 mm), which are hosts of A. ervi, A. pisum is the biggest (≤5.5 mm) (Blackman & Eastop, 2008).

Parasitoids with smaller wings emerged from aphid hosts feeding on cereals (wheat and oats), while from A. pisum feeding on legumes (alfalfa, clover and pea) the emerged individuals had larger wings. Although the effects of plant species on the A. ervi biotypes were not addressed here, this should not be completely neglected. Some evidences suggest that the preference of A. ervi biotypes toward plant/aphid host volatiles will eventually lead them to the adequate aphid host (Daza-Bustamante et al., 2002). Host and plant preferences could cause physiological changes in A. ervi as suggested by Cameron, Powell & Loxdale (1984). This could explain the variability in body size of parasitoids and the morphological differentiation of the forewings among the analyzed biotypes. The influence of host/plant association on morphological differentiation of forewings has been also shown in other studies of braconid wasps; e.g., biotypes from the genus Eubazus (Nees, 1814), a parasitoid of the conifer bark weevil (Villemant, Simbolotti & Kenis, 2007) or Lysiphlebus fabarum (Parreño et al., 2016).

Variations of the shape of insect wings are known to affect flight ability, which in turn could alter the host and mate allocation (Kölliker-Ott, Blows & Hoffmann, 2003). Betts & Wootton (1988) studied the effects of wing structure on the flight of six butterfly species and showed that there was a correlation between flight performance and wing shape. Additionally, studies have described how the wing shape can alter predation success by dragonflies (Combes, Crall & Mukherjee, 2010) and the ability of damselflies to avoid predation by passerine birds (Outomuro & Johansson, 2015). More specifically, parasitoids are affected by changes in wing size and shape. The wing size and shape of Trichogramma brassicae (Bezdenko, 1968) and Trichogramma pretiosum (Riley, 1879) as egg parasitoids, increase the ability to locate host eggs. Differences in wing size and shape were found between parasitoids obtained from field conditions compared to those parasitoids that were reared in the laboratory (Kölliker-Ott, Blows & Hoffmann, 2003). Authors suggest that wing shape and wing size can be reliable predictors of field fitness for these parasitoid species. In the present study, the biotypes of A. ervi emerged from A. pisum had larger and broader forewings compared to the other studied biotypes. These differences of wing shape and size could affect the fitness of A. ervi and its ability to find aphid hosts. Further research to determine the most suitable aphid host for A. ervi to increase its fitness will lead to enhanced rearing conditions for A. ervi and consequently, will improve any future inundative biological control strategies with this parasitoid.

Conclusion

Given the low genetic variability of Aphidius ervi in Chile, the main factor affecting morphological variations of A. ervi forewings is their aphid host. Forewing shape variability is partly influenced by allometric effects. The greatest difference in A. ervi wings among aphid hosts were observed between A. pisum and the cereal aphids in general.

Supplemental Information

Figure S1 Principal component analysis

Distribution of Aphidius ervi biotypes in the morphospace defined by PC1 and PC2 axes. The total variability explained for PC1 + PC2 = 37.39%.

Click here for additional data file.

Figure S2 Cannonical variate analysis

Distribution of Aphidius ervi biotypes in the morphospace defined by CV1 and CV2 axes. The total variability explained for CV 1 + CV 2 = 61.4%.

Click here for additional data file.

The authors wish to thank to Dr. Ana Ivanović (Faculty of Biology, University of Belgrade, Serbia) for the assistance in geometric morphometrics analyses.

Additional Information and Declarations

Competing Interests

Author Contributions

Data Availability

The authors declare there are no competing interests.

Cinthya M. Villegas conceived and designed the experiments, performed the experiments, analyzed the data, contributed reagents/materials/analysis tools, wrote the paper, prepared figures and/or tables, reviewed drafts of the paper.

Vladimir Žikić and Saša S. Stanković conceived and designed the experiments, analyzed the data, contributed reagents/materials/analysis tools, wrote the paper, prepared figures and/or tables, reviewed drafts of the paper.

Sebastián A. Ortiz-Martínez wrote the paper.

Ainara Peñalver-Cruz wrote the paper, reviewed drafts of the paper.

Blas Lavandero conceived and designed the experiments, analyzed the data, contributed reagents/materials/analysis tools, wrote the paper, reviewed drafts of the paper.

The following information was supplied regarding data availability:

Figshare: https://figshare.com/articles/Ervi_morphoj/4650271.

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
