# Peer review of "Morphological variation of Aphidius ervi Haliday (Hymenoptera: Braconidae) associated with different aphid hosts"

_PeerJ, doi:10.7717/peerj.3559_

## Round 0.1 · original submission · Major Revisions

Both reviewers consider this to be a solid paper, but state that it needs some substantial revisions. I am also suggesting major revisions. Much of the revision work revolves around correcting and tightening up the language. In addition, both reviewers state in their written review – and even more in their marked-up manuscripts which the co-authors will need to review carefully – the need for revision or response to a number of scientific concerns.

Reviewer 1 also would like to see "increased clarity around the statistical analyses used, the results of these analyses, and the conclusions". And I fully concur with that.

Thank you for submitting your MS to PeerJ, and we look forward to your revisions and rebuttal. As this is a recommendation of major revisions, I may elect to send it out for another round of review to the two reviewers – who I am thankful for graciously providing their expertise – if they are willing to do so.

·

Basic reporting

This paper was written clearly and concisely for the most part. A few grammatical corrections (edited in the pdf) will improve the paper. The abstract, introduction, and discussion sections were appropriate and clearly written. The methods section was clear; however, there should be a more detailed description of the statistical methods used in analyzing the data. Instead the stats were all mentioned for the first time in the results section, with little description of the intent of the various statistical methods. As well, the results section would be improved by including figures of all the multivariate analyses (PCA, CVA), even though there was no clear separation of groups. It would help the reader visualize the results of the analyses more clearly. More detailed comments are in the text.

Experimental design

The experimental design was clear and appropriate for the question being investigated. As mentioned above, there needs to be more clarity surrounding the statistical analyses and the goal of each statistical method.

Validity of the findings

The data seem to support the existing literature on A. ervi in Chile, and on the role of host/environment on wing shape. However I feel the authors are trying to find evidence for shape differences between biotypes, when in fact the data show limited differences (which is perhaps to be expected, based on the literature). I recommend that the authors examine their results carefully, and then re-work the discussion to ensure that their conclusions are in fact supported by the data. There is some confusion with regards to shape vs size, and the exact conclusions that can be drawn. The confusion may just be on the part of the reviewer, but in this case it’s important for the authors to clearly link the results and discussion sections. Please see more detailed notes in the attached review copy of the manuscript.

Additional comments

This paper addresses an interesting question, has a clear experimental design and I believe should ultimately be published. However, increased clarity around the statistical analyses used, the results of these analyses, and the conclusions that can be drawn is needed prior to Acceptance.

Reviewer 2 ·

Basic reporting

This manuscript needs a bit of work before publishing should be considered. There are numerous typos and grammatical errors throughout. I corrected some, but the manuscript really should have been gone through much more carefully before submitting to a journal.
I think the abstract should be re-worded. While the introductory letter to the journal/editor sounded well written and explained what was to be expected in the manuscript, I found the actual abstract to be somewhat difficult to follow.
The image file should be saved in some kind of commonly used format like a jpeg, tiff, png, etc. I am not familiar with this format and my computer would not open it.

Experimental design

No comment

Validity of the findings

No comment

Additional comments

(Line 54-57) Please re-word sentence and maybe split into 2 sentences. It is cumbersome and difficult to read.

(Line 58-59) This sounds contradictory. Do you mean great geographic range to find hosts in? Or that they are highly morphologically/physiologically specialized so they can attack a broad range of hosts?? Typically "highly specialized" does not fit with "great host range". you need to clarify/re-word.

(Line 75) The first time you use the name of your focus species in the main text, you need to spell out the authority’s name and the year described. The authority of the species should also be in the reference section with the rest of the citations.
You also need to say something here about this being a wasp, and what family & subfamily it is in. Stating it in the title is not sufficient.

(Line 77-81) The authority of these species names should also be spelled out in full with the date the first time they are used and put in the reference section.

(Line 73-75) Tone down the use of words like “obvious” throughout the manuscript. Whether or not it is obvious to you, you need to convince the reader that your assumption makes sense because of x, y, and supported by study z. It would read better if you said something like “this is likely due to reason____ as found previously in this species (reference)”.

Annotated reviews are not available for download in order to protect the identity of reviewers who chose to remain anonymous.

---

## Round 0.2 · Minor Revisions

Following a second review of the revised MS and associated rebuttal, the two initial referees have returned positive responses. I have also read through the revised MS and rebuttal and find the revision to be adequate in addressing the initial concerns.

There are still some minor grammatical issues, and both reviewers have pointed these out nicely in the marked-up PDFs that the authors will receive with this decision.

I also note that line 648-649 should read: “The biotypes that were used were Acyrthosiphon pisum from alfalfa…”

Please make these minor revisions and respond with a listing of the changes that you have made.

Thank you to the referees for their work on this article, and to the authors for an easy-to-process response.

·

Basic reporting

The writing and flow of the paper is much improved. I have made a few minor grammatical corrections/suggestions.

Experimental design

I appreciate the additional information regarding the statistical analyses. The analyses and results are now much easier to understand.

Validity of the findings

There is great improvement in linking the results/discussion to the stats and the literature.

Additional comments

I appreciate the extensive work the authors put into addressing the review comments. This paper reads much more clearly now, and is suitable for publication in PeerJ. I have made a number of minor changes throughout the text, but these are generally grammatical in nature.

Reviewer 2 ·

Basic reporting

Please see attachment for specific comments and suggestions.

Experimental design

Please see attachment for specific comments and suggestions.

Validity of the findings

Please see attachment for specific comments and suggestions.

Additional comments

Overall the manuscript is much improved since the original submission. Please see my text specific comments/suggestions in the attached file

Annotated reviews are not available for download in order to protect the identity of reviewers who chose to remain anonymous.

---

## Round 0.3 · accepted · Accept

Thank you to the reviewers (in two rounds) for their helpful comments and hard work that have substantially improved this MS. And thanks to the authors for working through the process as well.